# Unstimulated Parotid Saliva Sampling in Juvenile Idiopathic Arthritis and Healthy Controls: A Proof-of-Concept Study on Biomarkers

**DOI:** 10.3390/diagnostics10040251

**Published:** 2020-04-24

**Authors:** Alexandra Dimitrijevic Carlsson, Bijar Ghafouri, Carin Starkhammar Johansson, Per Alstergren

**Affiliations:** 1Center for Oral Rehabilitation, Department of Biomedical and Clinical Science, Linköping University, 581 83 Linköping, Sweden; carin.starkhammar.johansson@regionostergotland.se; 2Orofacial Pain Unit, Faculty of Odontology, Malmö University, Carl Gustafs väg 34, 214 21 Malmö, Sweden; per.alstergren@mau.se; 3Pain and Rehabilitation Centre, Department of Health, Medicine and Caring Sciences, 581 83 Linköping, Sweden; bijar.ghafouri@liu.se; 4Orofacial Pain and Jaw Function, Institute of Dental Medicine, Karolinska Institute, Alfred Nobels allé 8, 141 52 Huddinge, Sweden; 5Specialized Pain Rehabilitation, Skåne University Hospital, Lasarettsgatan 13, 222 41 Lund, Sweden

**Keywords:** biomarkers, children, juvenile idiopathic arthritis, parotid saliva, saliva collection

## Abstract

The aims of this proof-of-concept study were to develop a collecting method for unstimulated parotid saliva in juvenile idiopathic arthritis (JIA) patients and healthy children and to investigate if inflammatory biomarkers could be detected in these samples. Forty-five children with JIA (median age of 12 years and 25th–75th percentile of 10–15 years; 33 girls and 12 boys) and 16 healthy children as controls (median age of 13 years and 25–75th percentile of 10–13 years; 11 girls and 5 boys) were enrolled in this study. Unstimulated parotid saliva was collected with a modified Carlson–Crittenden collector. The salivary flow rate and salivary concentrations of total protein and inflammatory mediators were assessed. The Meso Scale Discovery electrochemiluminescence immunoassay was used for analyzing protein concentrations and the inflammatory biomarkers. Sufficient parotid saliva volumes to be analyzed could be collected with the collection device. JIA patients had a lower sampling saliva volume (*p* = 0.008) and saliva flow rate (*p* = 0.039) than controls. The total protein concentrations and inflammatory biomarkers were measured in the last six healthy subjects. The median protein concentration was 1312 µg/mL (25th percentile: 844 µg/mL and 75th percentile: 2062 µg/mL; *n* = 6) and quantifiable concentrations of 39 inflammatory proteins could be assessed in these samples. In conclusion, this study indicates that the saliva sampling method, as used in the present study, is able to collect sufficient sample volumes in children, and that it is possible to analyze various inflammatory biomarkers in the collected saliva.

## 1. Introduction

Juvenile idiopathic arthritis (JIA) is defined as arthritis of unknown origin with onset before the age of 16 and persisting for at least six weeks. JIA is categorized into seven disease subtypes [1]. The disease is characterized by autoimmune reactions often targeting synovial tissues, resulting in chronic arthritis [2,3]. A large proportion of these patients develop temporomandibular joint (TMJ) inflammation. Although TMJ pain is rare in JIA, untreated TMJ arthritis may lead to pain, but also cartilage and bone tissue destruction, growth disturbances, as well as functional and esthetic deformities [4,5,6]. Therefore, early identification of TMJ involvement is very important in order to enable the prevention of pain and tissue damage. Human saliva is a complex fluid and is rich in immunological components. Collection of saliva is a simple, non-invasive, and non-stressful procedure, in contrast to blood sampling. There seems to be a strong correlation between plasma and saliva content of warfarin [7], and saliva analysis has been suggested for monitoring treatment with warfarin [8]. Saliva sampling might, therefore, be an attractive alternative for diagnostic testing in children [9]. Advances in biotechnology have led to an interest in saliva diagnostics for monitoring disease [10]. Several types of inflammatory biomarkers associated with both oral diseases and inflammatory disease have been identified in adult saliva, for example, interleukins (IL-1, IL-6, and IL-8), tumor necrosis factor (TNF), and matrix metalloproteinases (MMP)-8 and MMP-9 [11,12]. Salivary biomarkers have been found to be useful in diagnosing children with metabolic disorders like obesity and diabetes [13]. For example, children with type 1 diabetes show higher levels of salivary pro-inflammatory biomarkers compared with healthy controls [14]. Moreover, biomarkers for predicting dental caries have been identified in children [15].

There are a number of methods for collecting whole saliva and glandular saliva where the methodology may impact the results regarding saliva composition. Sampling of stimulated or non-stimulated saliva influences the result regarding flow rates [16] and biomarkers [17]. Whole saliva contains a mix of secretions from salivary glands (parotid, submandibular, sublingual, labial, buccal, lingual, and palatal glands). It also contains bacteria, gingival cervical fluids, epithelial cells, food debris, and leucocytes [18]. These might interfere with protein identification because of their content of enzymes, which may degrade the substances before they can be detected. There are more enzymes in whole saliva than in parotid salivary secretions [19].

Whole saliva is also dominated by mucous cell secretions, which are rich in mucins and other glycoproteins. The mucins are high molecular weight polypeptides that stick together and have low solubility. Whole saliva usually requires centrifugation or filtration to remove mucins and cellular contaminants. Such centrifugation may also remove other proteins [20].

The parotid glands produce purely serous and watery saliva that is non-contaminated until entering the oral cavity, which could provide an advantage if it can be collected [21].

The aims of this proof-of-concept study were to develop a collecting method for unstimulated parotid saliva in juvenile idiopathic arthritis (JIA) patients and healthy children, and to investigate if inflammatory biomarkers could be detected in these samples.

## 2. Materials and Methods

### 2.1. Subjects

This study was conducted at the Centre of Oral Rehabilitation in Linköping, Sweden between 2015 and 2018. Forty-five JIA patients aged between 6 and 16 years (33 girls and 12 boys) were included (Table 1).

The JIA patients were referred consecutively from four pediatric departments in south-east Sweden (Linköping University Hospital, Vrinnevi Hospital/Norrköping, Motala Hospital, and Västervik Hospital). The inclusion criterion was a JIA-diagnosis according to the criteria of the International League of Association for Rheumatology (ILAR) [1]. The exclusion criteria were diabetes, inflammatory-bowel disease, and other chronic pain conditions than JIA. Nineteen (43%) of the patients had oligoarthritis, 15 (33%) had polyarthritis, and 11 (24%) had other subtypes of JIA (systemic arthritis, psoriatic arthritis, enthesitis-related arthritis, and undifferentiated arthritis). At inclusion, 34 patients were on anti-rheumatic therapy, 28 were on methotrexate, 12 were on biologics (adalimumab or etanercept), and 7 were on prednisolone. The patients without antirheumatic therapy were on NSAIDs (eight patients) or in remission. Sixteen healthy individuals (Table 1) with no pain from the orofacial region were recruited from the Public Dental Health Clinics in Linköping, Sweden. The exclusion criteria for the healthy individuals were rheumatic disease, diabetes, inflammatory-bowel disease, and other chronic pain conditions.

This methodological proof-of-concept study is part of a large longitudinal study of TMJ in JIA patients in relation to blood, saliva, periodontal, and radiographic biomarkers. There are no previous studies on inflammatory markers in saliva in JIA children, nor are there any high-quality studies on the prediction of TMJ involvement in JIA. A proper power calculation was thus difficult, also considering that there is only a very limited number of JIA patients available. We included all JIA patients referred to the Center for Oral Rehabilitation from the four hospitals.

### 2.2. Ethical Approval

This research was conducted in accordance with accepted ethical standards for research practice and was approved by the regional ethics committee in Linköping11-02-2015 (Dnr 2014/461–31). All subjects/parents received both verbal and written information about the study and signed an informed consent prior enrollment. They were also informed that they could cancel their participation in the study without any consequences for their normal care at the clinic and they had the possibility to contact the project manager for questions at any time during the study.

### 2.3. Parotid Saliva Collection Method

Unstimulated parotid gland saliva was collected intraorally using a modified Carlson–Crittenden collector [22] (Figure 1).

The device was manufactured by a technician at MedicinTeknik, Karolinska University Hospital, Huddinge, Sweden. A single examiner (A.D.C.) collected the saliva samples. The saliva collection was carried out between 08:30 and 11:30. The children were instructed to sit upright with the jaw in relaxed position and not to talk. During the procedure, the children were able to watch cartoons or use their mobile phone as distractions.

The Stensen’s duct was located, and the surrounding buccal mucosa dried with gauze. The collector was placed over the papilla of Stensen’s duct (Figure 1). Parotid saliva was collected during passive drooling via a 25 cm plastic tube into a 1.5 mL Eppendorf Tube (polypropylene tube) (Figure 1). Collection time varied from 5 to 60 min depending upon the time needed to collect an adequate sample volume. The sample volume was determined by the volume markings on the Eppedorph tubes. Total sampling time was recorded, and salivary flow was measured by dividing the collected saliva volume with the collection time (mL/min).

Immediately after sampling, the collected saliva was placed in a freezer (−30 °C) at the clinic. Later the same day, as soon as possible within a 4 h period of time, the samples were transported in a transport cooler bag to the laboratory to be stored at −70 °C until analysis. However, it has been reported that a saliva sample can be stored at room temperature for 30–90 min or at 4 °C for up to six hours to prevent degradation of salivary molecules [23].

In order to check for a systematic drift in operator-dependent sampling performance over time, the first and last ten samplings were compared. The first ten samples were obtained from nine patients and one healthy individual, whereas the last ten samples were from two patients and eight healthy individuals.

### 2.4. Biochemical Analysis

The concentrations of inflammatory proteins in saliva samples were analyzed from the last six healthy control subjects from the original group of 16 controls with the purpose to investigate the possibility to detect these proteins in the samples. The samples were analyzed using a commercially available V-PLEX human biomarkers (Meso Scale Diagnostics, Rockville, MD, USA) multiplex assay. The assay contained 39 biomarkers (see Table 4) and was based on an electrochemiluminescent detection method and performed exactly according to the manufacturer’s recommendations. Data were collected and analyzed using MESO QUICKPLEX SQ 120 instrument equipped with DISCOVERY WORKBENCH^®^ data analysis software (Meso Scale Diagnostics, Rockville, MD, USA).

The precisions based on both intra- and interassay coefficient of variations were <10% within the detection limits. The detection limit for each investigated biomarker is shown in Table 4 [24,25].

Total protein concentration was measured using 2D Quant-kit (GE Healthcare Biosciences AB, Uppsala, Sweden), according to the manufacturer’s protocol.

### 2.5. Statistical Analysis

Non-parametric statistics were used throughout the study owing to the skewed distribution of some of the variables and, regarding the biomarker analysis, owing to the low number of observations. For descriptive statistics, the median as well as the 25th and 75th percentiles were reported. For analytical statistics, the Mann–Whitney U-test was used to calculate the significant differences with respect to the time of collection, volume collected, and flow rate between groups (patients vs. healthy individuals) or between the ten first or last samplings. A probability level of *p* < 0.05 was considered as significant. IBM SPSS Statistics 25 was used for statistical analyses.

## 3. Results

The results of the sampling time, sample volume, and flow rate are shown in Table 2.

A significantly lower saliva flow rate was found for the JIA patients compared with the healthy children (*p* = 0.039). There was no significant relation between the saliva flow rate and age or gender. There was no significant difference regarding the saliva volume, saliva flow rate, or sampling time between the first ten samplings and the ten last samplings (Table 3).

Regarding the six healthy individuals in which we analyzed the inflammatory biomarkers, the median (25th/75th percentiles) total protein concentration was 1312 (843/2062) µg/mL. Table 4 shows the parotid saliva concentrations of 39 mediators in the six healthy individuals, as well as the proportion of samples where the respective mediator could be detected above the detection limit of the assay.

## 4. Discussion

The main findings of this proof-of-concept study can be summarized as follows: the saliva collecting method tested in this study can be used to sample unstimulated parotid saliva with sufficient volume to analyze in children with JIA and healthy children. Moreover, the combination of this saliva sampling method and the MESO multiplex bioassay seems to be able to detect various inflammatory biomarkers in the saliva samples from healthy children.

The method described in the present study managed to regularly collect sufficient saliva volume to be assayed by the multiplex assay. Using the assay, we managed to detect and quantify the chosen 39 mediators in the saliva from healthy individuals. For most mediators, we managed to detect the mediator in every sample. For seven of the mediators, the method could not detect the mediator in every sample. The method must thus be considered efficient to collect relevant parotid saliva samples.

Our study found a lower unstimulated parotid saliva flow rate in the patients with JIA compared with healthy individuals. The difference was not substantial, but still significant. To the best of our knowledge, unstimulated parotid saliva has not been measured in JIA patients before. The lower salivary flow rate in JIA patents may be explained by the higher degree of psychological distress [26] and medication [27] found in JIA patients, which may very well affect the salivary gland function and composition of saliva. The salivary flow rate increases with age and boys have higher flow rates than girls. The salivary glands seem to be fully developed at the age of 15 [28]. In our study, most subjects were younger than 15, the proportion of girls was higher, and 45 of the 61 subjects had an autoimmune disease, which may explain the lower flow rates found in our study.

Regarding flow rate, resting parotid saliva flow rate was investigated in another study in 207 healthy children of both sexes between 3 and 16 years of age. They found a mean flow rate for all 207 subjects of 0.034 mL/min. In that study, the flow rate differed between the younger children (3–8 years) and the older children (9–16 years) [29]. The flow rates in our study were surprisingly lower than what was found in that study. One explanation might be that, in our study, 45 patient had an autoimmune disease and their medications may also be associated with a lower saliva flow rate [30]. In adults, autoimmune disease may cause inflammation of the salivary glands and associated reduction in salivary flow rate, as well as alterations in saliva composition [31,32]. Juvenile Sjögren’s syndrome is rare and possibly underdiagnosed. Swelling of the major salivary glands and hyposalivation are common in these patients [33]. This might be a factor that contributes to the lower salivary flow rate found in JIA patients. The time it took to collect sufficient saliva volume was, however, about the same in the patients and healthy subjects. Together, this indicates that the reason for lower saliva flow rates in our sample is owing to the JIA patients included in our study.

The various mediators could be detected in most of the salivary samples with the assay used in this study. Of the 39 investigated mediators, seven were not detected in every sample. This is not possible to explain today, especially because the samples where these mediators were detected were well over the detection limit. This means that investigation of the mediators in the present study in relation to clinical, radiological, and laboratory variables in JIA patients in order to increase knowledge about the disease itself and the potential value of saliva analysis in JIA is feasible and of great interest. Certainly, every mediator must be considered separately regarding feasibility.

### Methodological Considerations

Parotid saliva collection is time-consuming and requires a special device, the parotid cup. The method is operator-dependent to some degree because it takes some time to learn the technique [17]. The cups are not commercially available, but must be manufactured in a medical technique workshop. Before the start of the study, we tested a prototype, but discovered that the metal tube was too narrow and took too long time to collect saliva. We redesigned the cup by adding a wider metal/silicone tube. After that, collection of saliva worked much better by being quicker and more stable.

Regarding obtaining saliva samples, it is important to allow enough time for the saliva to be collected. During the collection, it is important to not move the cup because that will disturb the collection. The subject needs to sit still. In this study, this worked well for children aged six years and older. However, it may not work for younger children.

There is most likely variability of the sampling procedure. We aimed for a certain sample volume, and the sampling procedure thus varied between subjects owing to differences in saliva flow. The time differences may change the saliva composition, but, from an ethical point of view, we could not ask the children with a high flow rate to perform the sampling for a longer time than needed for the sampling volume. As a proof-of-concept study, we do not consider this to influence our results.

A possible interaction between the sampling cup material and the biomarkers was out of the scope of investigation for this study, but needs to be addressed in future studies comprising biomarker concentrations.

## 5. Conclusions

This proof-of-concept study indicates that the saliva collecting method tested in this study can be used to sample unstimulated parotid saliva with sufficient volume to analyze in children with JIA and healthy children. Moreover, the combination of this saliva sampling method and the MESO multiplex bioassay seems to be able to detect various inflammatory biomarkers in the saliva samples from healthy children.

## Figures and Tables

**Figure 1 diagnostics-10-00251-f001:**
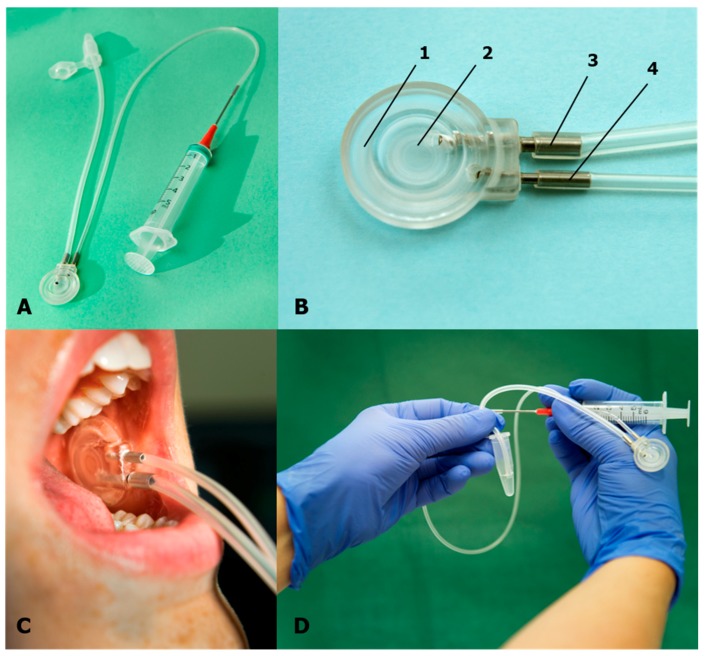
(**A**) Parotid saliva collection set up (cup, metal and silicone tubes, syringe, and cannula). (**B**) Parotid saliva cup collection device: 1: outer cup for fixation of the cup to the mucosa around the Stensen´s duct by vacuum; 2: inner cup inclosing the opening of the Stensen´s duct; 3: metal tube with silicone tubing and outlet tube for the saliva; and 4: tube for the production of a vacuum in cup. (**C**) Cup placed over the papilla of the Stensen´s duct for parotid saliva collection. (**D**) Saliva collected in an Eppendorf tube. Photo: S. Gustavsson, Redakta Reportage, Linköping, Sweden.

**Table 1 diagnostics-10-00251-t001:** Demographic data 45 patients with juvenile idiopathic arthritis and 16 age- and sex-matched healthy individuals.

		Patients				Healthy Individuals			
			Percentiles				Percentiles		
		Median	25th	75th	*n*	Median	25th	75th	*n*
Individuals									
Age	years	12	10	15	45	13	10	13	16
Sex	boys/girls				12/33				5/11
Age at diagnosis	years	9	5	12	45	n.a			
Disease duration	years	4	3	7	45	n.a			

*n* = number of observations.

**Table 2 diagnostics-10-00251-t002:** Unstimulated parotid saliva sampling time and saliva flow rate in 45 patients with juvenile idiopathic arthritis and 16 healthy sex- and age-matched healthy individuals.

		Patients				Healthy Individuals				
			Percentiles				Percentiles			
		Median	25th	75th	*n*	Median	25th	75th	*n*	*p*
Time	min	15	12	30	45	14	10	19	16	0.162
Flow	mL/min	0.016	0.007	0.020	45	0.020	0.014	0.036	16	0.039

**Table 3 diagnostics-10-00251-t003:** Unstimulated parotid saliva sampling time and saliva flow in 45 patients with juvenile idiopathic arthritis and 16 healthy sex- and age-matched healthy individuals comparing the ten first and last sample occasions.

			First 10				Last 10			
			Percentiles				Percentiles			
Saliva		Median	25th	75th	*n*	Median	25th	75th	*n*	*p*
Time	min	14	11	15	10	16	13	22	10	0.218
Flow	ml/min	0.025	0.020	0.030	10	0.020	0.010	0.020	10	0.165

**Table 4 diagnostics-10-00251-t004:** Concentrations of 39 inflammatory proteins in parotid saliva from six healthy children.

Abbreviations	Proteins	Detected in Proportion of Assayed Samples	Detection Limit (pg/mL)
bFGF	Basic fibroblast growth factor	6/6	0.14
CRP	C-Reactive Protein	5/6	2.9
Eotaxin	Eosinophil chemotactic protein	5/5	0.88
Eotaxin-3	Eosinophil chemotactic protein 3	5/5	1.7
Flt-1	Vascular endothelial growth factor receptor 1	6/6	1.2
GM-CSF	Granulocyte-macrophage colony-stimulating factor	5/5	0.11
ICAM-1	Intercellular adhesion molecule 1	6/6	2.55
IFN-γ	Interferon gamma	6/6	0.40
IL-10	Interleukin 10	6/6	0.03
IL-12/23p40	23p40 Interleukin 12/23 disulfide-linked p40	5/5	0.24
IL-12p70	Interleukin 12, p70	2/6	0.04
IL-13	Interleukin 13	6/6	0.73
IL-15	Interleukin 15	5/5	0.17
IL-16	Interleukin 16	5/5	0.80
IL-17A	Interleukin-17A	5/5	0.22
IL-1α	Interleukin 1α	5/5	0.04
IL-1β	Interleukin 1β	6/6	0.02
IL-2	Interleukin 2	6/6	0.05
IL-4	Interleukin 4	4/6	0.01
IL-6	Interleukin 6	6/6	0.04
IL-7	Interleukin 7	5/5	0.08
IL-8	Interleukin 8	6/6	0.03
IL-8(HA)	Interleukin anti-hu-8	2/5	186
IP-10	kDa interferon gamma-induced protein	5/5	0.09
MCP-1	Monocyte chemoattractant protein	5/5	0.07
MCP-4	Monocyte chemoattractant protein 4	5/5	1.40
MDC	Macrophage-derived chemokine	5/5	1.77
MIP-1α	Macrophage inflammatory protein 1-alpha	5/5	2.10
MIP-1β	Macrophage inflammatory protein 1-beta	5/5	0.04
PlGF	Placental growth factor	6/6	0.14
SAA	Serum amyloid A-1 protein	6/6	18
TARC	Thymus and activation-regulated chemokine	5/5	0.05
Tie-2	Angiopoietin-1 receptor/Endothelial tyrosine kinase	2/6	43
TNF	Tumor necrosis factor	6/6	0.06
VCAM-1	Vascular cell adhesion protein 1	5/6	7.8
VEGF	Vascular Endothelial Growth Factor	5/5	0.70
VEGF	Vascular endothelial growth factor	5/6	0.42
VEGF-C	Vascular endothelial growth factor C	6/6	18
VEGF-D	Vascular endothelial growth factor D	5/6	4.3

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
