# Peer review of "Unstimulated Parotid Saliva Sampling in Juvenile Idiopathic Arthritis and Healthy Controls: A Proof-of-Concept Study on Biomarkers"

_diagnostics, 2020, doi:10.3390/diagnostics10040251_

Round 1
Reviewer 1 Report
COMMENTS TO THE AUTHOR(S)
The manuscript requires major revisions in order to improve the language level as well as to better explain the experimental results, which do not allow to confirm the initial hypothesis of the work, i.e. “… investigate assessment of inflammatory biomarkers that could be serve as reference values for subsequent investigations in JIA.” In my opinion, the current version of the manuscript does not show the required quality to be published in Diagnostics.
General comment:
The main idea of the authors is interesting because they tried to use a panel of salivary proteins as potential biomarkers to stratify JIA patients. Unfortunately, experimental data did not support their initial idea since they missed the following aspects:
- authors did not determine the salivary proteins in all patients, failing with the main goal of the article. I suggest to include all the experimental results;
- authors should justify the number of enrolled patients/control by performing a power analysis;
- authors should discuss the variability of the sampling procedure. In fact, it is well-known in literature that the sampling procedure may alter the chemical composition of saliva thus limiting the spread of saliva analysis in the clinical setting. I suggest to discuss all these points in the article;
- in addition, is not clear if the authors developed and validated the analytical procedure or they used an already published work. Anyway, in both cases an in-house validation must be carried out. So, the authors should include all the validation parameters (e.g. recovery, precision, matrix effect and etc) in the paper.
Specific comments:
L22-25, please modify the sentence in order to clarify the concept.
L29-30, why the panel of salivary biomarkers was determined only in six subjects? Pleas, see the general comment above.
L48-62, I suggest to extend the introduction part of the work by including additional references with the aim to better explain the need of using standardized collection procedures in the field of saliva analysis. Moreover, I suggest to explain the main application of saliva analysis. The following articles can be useful for the authors and should be included as references.
doi.org/10.1016/j.microc.2017.02.032
doi.org/10.1016/j.microc.2017.04.033
doi.org/10.1371/journal.pone.0114430
doi.org/10.1016/j.trac.2019.115781
doi.org/10.1016/j.microc.2017.02.010
L74, please explain that the p value is referred to the comparison between the two medians. In addition, I suggest to correct the unit of measure of volume and to show the volume in uL instead of mL. In this way, the increased number of digit numbers allows to clearer the difference between these two median values. Why did you used a different sampling time to collect saliva?
L79, why did you compared the first-10 and last-10 samples? What’s the rationale behind this comparison?
L82, why did you determine the total protein content only for 6 healthy participants? Please include the total protein content for all the enrolled volunteers/patients. In some case, the protein was determined only in two samples (IL-8(HA) and Tie-2) and therefore I suggest a different way to express the experimental values. In addition, I suggest to modify the digit numbers according to the method variability.
L86, please include the limits of detection.
L89-90, please modify the sentence with the aim to better clarify the concept.
L105, median is an exact value and thus should be not expressed as a range.
L112, did you evaluate the impact of flow rate on the proteins concentration in saliva?
L120, did you test the possible interaction between proteins and parotid cup?
L165-166, the use of an external stimuli may modify the unstimulated flow rate affecting the accuracy of the flow rate value. Did you evaluate the impact of watching cartoons on the flow rate value?
L170, I suggest to include that the flow rate was estimated by considering the density of saliva equal to 1 g/mL.
L172-173, the storage conditions may affect the structural conformation of proteins. For this reason it is very recommended to not changes the storage temperature by time. Did you evaluate the stability of your target analytes? What this the difference in time between the collection and storage and between storage and analysis? These information must be included in the text.
L174, I suggest to explain if the method was already validated for saliva analysis or not. Anyway, authors should perform a an in-house validation and reports experimental results (experimental variability, matrix effect, recovery, LOD and etc).
L187, I suggest to check the statistical analysis because in some case the target analyte was determined in only two sample. In such cases, talking of “non-parametric statistics” is not really appropriate.
L191, I suggest to rewrite the conclusions considering all the experimental data.
Author Response
Answer to Review 1
Thank you for your thoughtful and valuable suggestions in order to improve our manuscript. We have accepted your suggestions and changed the manuscript based on these. See below for detailed answers and comments.
General comments
- We have clarified the Aims of this proof-of-concept study. We did not include all samples in the MESO bioassay since the aim was solely to test if these inflammatory biomarkers could be detected in our samples. For that we considered six samples to be sufficient. We have also changed the table accordingly.
- This methodological proof-of-concept study is part of a large longitudinal study of temporomandibular joint involvement in JIA patients in relation to blood, saliva, periodontal and radiographic biomarkers. The power analysis was primarily performed for those aspects. In addition, there are no previous studies on inflammatory markers in saliva in JIA children, nor are there any high-quality studies on prediction of TMJ involvement in JIA. A proper power calculation was therefore difficult, also considering that there are only a number of JIA patients available. We included all JIA patients referred to the Center for Oral Rehabilitation from four pediatric department in south-east Sweden (Linköping University Hospital, Vrinnevi Hospital/Norrköping, Motala Hospital and Västervik Hospital). For this study 45 patients and 16 healthy age- and sex-matched control individuals from the Public Dental Health Clinics in Linköping, Sweden was possible to include. In our ethical permission, 70 patients and 70 healthy were planned for. We have added this to the Materials and Methods section.
- There is most likely variability of sampling procedure. We aimed for a certain sample volume, thus the sampling procedure varied between subjects due to differences in saliva flow. The time differences may change the saliva composition but, from an ethical point-of-view, we could not ask the children with high flow rate to perform the sampling for a longer time than needed for the sampling volume. As a proof-of-concept study, we do not consider this to influence our conclusions, i.e. the method allows parotid saliva sampling and inflammatory biomarker are possible to detect in these samples. We have clarified this in the Discussion.
- The MESO bioassay was performed according to the manufacturer’s specification and instruction. The precision based on both intra and inter-assays variations were <10% within the detection limits, for more detail about the analytical parameters see Supplementary table 1.
Changes in inflammatory plasma proteins from patients with chronic pain associated with treatment in an interdisciplinary multimodal rehabilitation program – an explorative multivariate pilot study. Björn Gerdle, Emmanuel Bäckryd, Torkel Falkenberg, Erik Lundström and Bijar Ghafouri. Scand J Pain 2019 https://doi.org/10.1515/sjpain-2019-0088.
Validation and comparison of two multiplex technologies, Luminex® and Mesoscale Discovery, for human cytokine profiling. Ferdousi Chowdhury, Anthony Williams, Peter Johnson, Journal of Immunological Methods, Volume 340, Issue 1, 1 January 2009, Pages 55-64. Multiplex measurement of proinflammatory cytokines in human serum: Comparison of the Meso Scale Discovery electrochemiluminescence assay and the Cytometric Bead Array. Djeneba Dabitao, Joseph B. Margolick, Joseph Lopez, Jay H. Bream, Journal of Immunological Methods Volume 372, Issues 1–2, 30 September 2011, Pages 71-77
Specific comments
L22-25 We have changed the Abstract to clarify the concept, please also note that we have clarified the Aims accordingly
L29-30 We analyzed biomarkers in the six last healthy subjects included in order to investigate if inflammatory biomarkers were possible to detect with our assay, as proof-of-concept for the combination of the sampling method and assay. We have tried to clarify this in the manuscript and also removed the quantitative data for the biomarkers in the Table.
L48-62 We have expanded the Introduction according to the referee’s comment and included four of the suggested papers.
L74 We collected saliva until there was sufficient sample volume to be assayed. We have changed Tables 1 and 2 by removing “Volume” since this was the criteria for terminating the sampling, i.e. not interesting to compare between the groups. The common unit for measuring saliva flow is mL/min, why we chose to keep mL/min in the tables.
L79 This was performed in order to investigate if there was any systematic change over time, i.e. if the method is operator-dependent. We have clarified this in the Materials and Methods section.
L82 As mentioned above and due to the proof-of-concept design of this study, we analyzed the biomarkers only in the last six samples from healthy individuals in order to investigate if these included biomarkers could be detected. We did not have any aspiration to establish reference values in the present study, which we now understand that the first manuscript may have indicated unintentionally. We have updated the text and Table 4 accordingly. Please note that we assayed 5 or 6 samples for each biomarker but IL-8 and Tie-2 was only detectable in 2 of these samples.
L86 The limits of detection, expressed in pg/mL, for all substances are stated in table 4 in the revised manuscript.
L 89-90 We have changed the first sentence to “The main findings of this proof-of-concept study can be summarized as follows: the saliva collecting method tested in this study can be used to sample unstimulated parotid saliva with sufficient volume to analyze in children with JIA and healthy children. Also, the combination of this saliva sampling method and the MESO multiplex bioassay seems to be able to detect various inflammatory biomarkers in the saliva samples from healthy children.”
L105 We have changed the Discussion to “The salivary glands seems to be fully developed at the age of 15 [20]. In our study, most subjects were younger that 15, the proportion of girls was higher, and 45 of the 61 subjects had an autoimmune disease, which may explain the lower flow rates found in our study.”
L112 We only have data on total protein concentration for the six healthy individuals, unfortunately. A correlation between flow rate and total protein concentration in all samples would have been very interesting but for the moment it is not possible.
L120 No, we did not. Such interaction is possible and needs to be determined in the future when we will publish the biomarker concentrations for all included subjects. We added “A possible interaction between the sampling cup material and the biomarkers was out of the scope for this study to investigate but needs to be addressed in future studies comprising biomarker concentrations.” to the Materials and Methods section.
L165-166 Watching cartoons were used only for the youngest children to relax, pacify and comply. We did not analyze any influence of this stimuli on saliva flow rate or sampling time, given the proof-of-concept nature of this study, but such influence can’t be excluded.
L170 Sorry but we don’t understand this remark. We measured saliva flow rate in mL/min, which is not dependent on the saliva density. Maybe you mean something related to the total protein concentration? Please clarify in that case.
L172-173 We have rephrased to clarify: “Immediately after sampling, the collected saliva was placed in a freezer (-30°C) at the clinic. Later the same day, as soon as possible within a 4 hours period of time, the samples were transported in a transport cooler bag to the laboratory to be stored at -70°C until analysis. However, it has been reported that saliva sample can be stored at room temperature 30-90 minutes or 4 ºC up to 6 h to prevent degradation of salivary molecules (Chiappin S, Antonelli G, Gatti R, De Palo EF. Saliva specimen: a new laboratory tool for diagnostic and basic investigation. Clinica chimica acta; international journal of clinical chemistry. 2007;383:30–40.)
L174 Please, see the response to point 4. We added this in the manuscript under Biochemical analysis.
L187 We have removed the biomarker concentrations and only report the proportion of assayed samples where we could detect the separate biomarkers (Table 4). Therefore, no statistics are applied on this data.
L191 We have changed the Conclusion to: “This proof-of-concept study indicates that the saliva collecting method tested in this study can be used to sample unstimulated parotid saliva with sufficient volume to analyze in children with JIA and healthy children. Also, the combination of this saliva sampling method and the MESO multiplex bioassay seems to be able to detect various inflammatory biomarkers in the saliva samples from healthy children.”
Reviewer 2 Report
If I have understood well the aims of this study, the first was to evaluate how a modified collecting method of unstimulated parotid saliva works in juvenile idiopathic arthritis (JIA) and comparing the results of saliva collected with those from healthy children; and the second was to determine what inflammatory analytes could serve as possible inflammatory biomarkers in JIA.
However, according to how the study is outlined, only the first aim was achieved. I mean: I did not understand why the authors used only six healthy children to evaluate the 40 inflammatory biomarkers. If the aim was to obtain the reference intervals (RIs), it would be necessary a large population of healthy children according to, for example, the guidelines of the American Society for Veterinary Clinical Pathology (ASVCP 2011 guides) (1). But it would be more interesting to evaluate the salivary analytes selected in the whole population (45 children with JIA and 16 healthy children) to then compare results between groups. But always later of a previous analytical validation, at least a partial validation (2): reproducibility (Intra-assay or inter-assay precision) and accuracy (Lower Limit of Quantification, LLOQ; and linearity under dilution, LD). As the manuscript is written, the authors cannot conclude that a large number of inflammatory biomarkers can be detected in saliva by the analytical method used. Therefore, this manuscript needs a deep change to achieve the aims that authors want to pursue. After that, it would be a pleasure to review this manuscript again since the aims are novel.
Author Response
Answer to Review 2
Thank you for your thoughtful and valuable suggestions to improve our manuscript. We have changed the manuscript based on your recommendations.
- We have clarified the Aims of this proof-of-concept study. We did not include all samples in the MESO bioassay since the aim was solely to test if these inflammatory biomarkers could be detected (or not) in our samples. For that we considered six samples to be sufficient. We have also changed the table accordingly.
- First, we have revised the aim in order to clarify what we intended to do. We have also added data regarding reproducibility and accuracy in a supplementary table. We also added clarifying text under “Biochemical analysis” in the Materials and Methods section, together with two new references: i) Changes in inflammatory plasma proteins from patients with chronic pain associated with treatment in an interdisciplinary multimodal rehabilitation program – an explorative multivariate pilot study. Björn Gerdle, Emmanuel Bäckryd, Torkel Falkenberg, Erik Lundström and Bijar Ghafouri. Scand J Pain 2019 https://doi.org/10.1515/sjpain-2019-0088 and ii) Multiplex measurement of proinflammatory cytokines in human serum: Comparison of the Meso Scale Discovery electrochemiluminescence assay and the Cytometric Bead Array. Djeneba Dabitao, Joseph B. Margolick, Joseph Lopez, Jay H. Bream, Journal of Immunological Methods Volume 372, Issues 1–2, 30 September 2011, Pages 71-77.
Reviewer 3 Report
The authors present a new methodological approach to studying saliva biochemistry in rheumatic patients. Sjögren synfrome is often linked to rheumtaic disease. The authors can discuss this issue in the light of their approach and results.
Author Response
Answer to Review 3
Thank you for your suggestion to discuss the relation regarding Sjögren´s Syndrome linked to rheumatic disease and salivary flow. This is now added to the discussion section.
In adults, autoimmune disease may cause inflammation of the salivary glands and associated reduction in salivary flow rate as well as alterations in saliva composition (Helenius LM ACTA 2005, K Moen oral disease 2005). Juvenile Sjögren’s syndrome is rare and possibly underdiagnosed. Swelling of the major salivary glands and hyposalivation are common in these patients (ref Juvenile Sjögren’s Syndrome: Clinical Characteristics with Focus on Salivary Gland Ultrasonography. Daniel S. Hammenfors, Arthritis Care & Research Vol. 72, No. 1, January 2020, pp 78–87). This might be a factor that contributes to the lower salivary flow rate found in the JIA patients. The time it took to collect sufficient saliva volume was, however, about the same in the patients and healthy subjects. Together, this indicates that the reason for lower saliva flow rates in our sample is due to the JIA patients included in our study.
Round 2
Reviewer 1 Report
Dear Authors, the revision process significantly improved the quality of the manuscript, thus I suggest to accept the paper in the present form.
Regards.
Author Response
Thanks you for your thoughtful and valuable suggestions in order to improve our manuscript.
Reviewer 2 Report
Abstract
Lines 23-24: Please, consider to rewrite the sentence as: “Forty-five children with JIA (median age 12 years and 25th-75th percentile 10-15 years, 33 girls and 12 boys) and 16 healthy children as controls (median age 13 years and 25th-75th percentile 10-13 years, 11 girls and 5 boys) were enrolled in this study.”
Line 27: You measured protein concentrations and also the inflammatory biomarkers with the electrochemiluminescence immunoassay. That’s correct?
Line 35: Write a comma after “children”
Line 36: Authors need to specify what kind of biomarkers they have analyzed. For example: “(…) possible to analyze various inflammatory biomarkers in the collected saliva.”
Introduction
Line 41: Change “in to” for “into”.
Lines 43-47: Considering to modify this sentence like the following: “Although TMJ pain is how are rare in JIA, untreated TMJ arthritis may lead to pain, but also cartilage and bone tissue destruction, growth disturbances, as well as functional and esthetic deformities [4,6]. Therefore, early identification of TMJ involvement is very important in order to enable the prevention of pain and tissue damage.
Material and Methods
Lines 161-162: Do not repeat information, since the demographic data has been included in Table 5.
Lines 171-172: Please, remove “;” and change it by a comma.
Lines 173-174: Please, consider rewrite as following: “Sixteen healthy individuals (Table 5) with no pain from the orofacial region (…)”.
Lines 197-201: Please, consider replacing the paragraph with the following one: “The Stensen’s duct was located, and the surrounding buccal mucosa dried with gauze. The Collector was placed over the papilla of Stensen’s duct (Fig. 1). Collection time varied from 5 to 60 min depending upon the time needed to collect an adequate sample volume. During the procedure, the children were able to watch cartoons or use their mobile phone as distractions.”
Line 203: How did you calculate the volume of collected saliva? Weighing several empty Eppendorfs, and its average subtracts it with one with the saliva obtained? Or approximately with the value of the volume marked in the Eppendorf’s lateral? Please note that the volume value provided by the Eppendorf is only an estimate.
Line 217: The authors can refer to Table 4 to specify which biomarkers they refer to.
Lines 222-223: Check the word “pERformed”. I do not understand the sentence “(…) and the operator of the assay was, blinded to if the sample was from a patient or healthy individual”. I advise removing it.
Lines 224-225. Please, remove this sentence since you wrote it in line 219: “The MESO bioassay was performed according to the manufacturer’s specification and instruction.”. Change the sentence like the following: “The precisions based on both intra- and inter assay coefficient of variations (CV) were (…)”
Lines 226 and 227: Please, write a point after the word “limits”. Write the following: “For more detail about the analytical validation from the inflammatory parameters measured, see the supplementary table 1 [32.33].”
Lines 229-231: This sentence could go better in a Discussion paragraph as a limitation of the study.
Line 233: Give information about the intra and inter-assay coefficient of variation results, and detection limits offered by the manufacturer.
Line 235: Did you assess the normality distribution? Or it was decided because there was a big difference between both groups, the JIA and healthy group? IN any case, you must justify adequately why you used a non-parametric test. Change “25th/75th” for “25th-75th.
Lines 236-237: Complete the sentence: “For analytical statistics, the Mann-Whitney U-test was used to calculate the significant differences with respect to the time of collection, volume collected and flow rate between groups (patients vs. healthy individuals) or between the ten first or last samplings.”
Results
Table 1. The units of volume are mL and not mL/min.
Line 86: How did you calculate it? You need to specify the kind of statistic calculation in the “Statistical analysis” sub-heading.
Lines 78-79: Those first and last ten samplings are only from the JIA group or form the healthy group or from both?
Table 3. There is no point in offering the results of time collection, volume, and flow rate from these six healthy controls since these data must be as the previous one. You can describe only the results from the protein concentration.
Line 93: Were not 40 inflammatory biomarkers according to the information offered in the “Material and Methods” heading?
Discussion
Lines 104-109: Please, describe which and which not inflammatory mediators can be detected in every sample.
Please, provide a paragraph of limitations. For example, no power study to determine if the statistic study had a type 2 error was performed, a higher proportion of girls than boys in both groups; and ideally, it would be desired also to assessing of the inflammatory mediators included in this study in saliva from the JIA patients.
Author Response
Answers to Review 2 second round
A warm and greateful thanks for your thoughtful and valuable suggestions in order to improve our manuscript. We have accepted your suggestions and changed the manuscript based on these. See below for detailed answers and comments.
Abstract
Lines 23-24: Please, consider to rewrite the sentence as: “Forty-five children with JIA (median age 12 years and 25th-75th percentile 10-15 years, 33 girls and 12 boys) and 16 healthy children as controls (median age 13 years and 25th-75th percentile 10-13 years, 11 girls and 5 boys) were enrolled in this study.”
Changed accordingly
Line 27: You measured protein concentrations and also the inflammatory biomarkers with the electrochemiluminescence immunoassay. That’s correct?
That is correct and has been changed in the manuscript.
Line 35: Write a comma after “children”
Done
Line 36: Authors need to specify what kind of biomarkers they have analyzed. For example: “(…) possible to analyze various inflammatory biomarkers in the collected saliva.”
We added “inflammatory”
Introduction
Line 41: Change “in to” for “into”.
Changed
Lines 43-47: Considering to modify this sentence like the following: “Although TMJ pain is how are rare in JIA, untreated TMJ arthritis may lead to pain, but also cartilage and bone tissue destruction, growth disturbances, as well as functional and esthetic deformities [4,6]. Therefore, early identification of TMJ involvement is very important in order to enable the prevention of pain and tissue damage.
Changed to “Although TMJ pain is rare in JIA, untreated TMJ arthritis may lead to pain, but also cartilage and bone tissue destruction, growth disturbances, as well as functional and esthetic deformities [4-6]. Therefore, early identification of TMJ involvement is very important in order to enable the prevention of pain and tissue damage”
Material and Methods
Lines 161-162: Do not repeat information, since the demographic data has been included in Table 5.
We’re sorry but as far as we can see, there is no double-reporting of data found in Table 5 in the text. (The number of the table is now, in the revised manuscript, Table 4, since we removed the former Table 3).
Lines 171-172: Please, remove “;” and change it by a comma.
Changed accordingly
Lines 173-174: Please, consider rewrite as following: “Sixteen healthy individuals (Table 5) with no pain from the orofacial region (…)”.
We changed this according to your suggestions.
Lines 197-201: Please, consider replacing the paragraph with the following one: “The Stensen’s duct was located, and the surrounding buccal mucosa dried with gauze. The Collector was placed over the papilla of Stensen’s duct (Fig. 1). Collection time varied from 5 to 60 min depending upon the time needed to collect an adequate sample volume. During the procedure, the children were able to watch cartoons or use their mobile phone as distractions.”
We acknowledge the need for a more logical order in the text here. We have rewritten the paragraph as follows, into three separate paragraphs:
“The device was manufactured by a technician at MedicinTeknik, Karolinska University Hospital, Huddinge, Sweden. A single examiner (ADC) collected the saliva samples. The saliva collection was carried out between 08.30 and 11.30 a.m. The children were instructed to sit upright with the jaw in relaxed position and not to talk. During the procedure, the children were able to watch cartoons or use their mobile phone as distractions.
The Stensen’s duct was located, and the surrounding buccal mucosa dried with gauze. The Collector was placed over the papilla of Stensen’s duct (Fig. 1). Parotid saliva was collected during passive drooling via a 25 cm plastic tube in to a 1.5 ml Eppendorf Tube (polypropylene tube) (Fig. 1). Collection time varied from 5 to 60 min depending upon the time needed to collect an adequate sample volume. Total sampling time was recorded, and salivary flow was measured by dividing the collected saliva volume with the collection time (mL/min).
Immediately after sampling, the collected saliva was placed in a freezer (-30°C) at the clinic. Later the same day, as soon as possible within a 4 hours period of time, the samples were transported in a transport cooler bag to the laboratory to be stored at -70°C until analysis. However, it has been reported that saliva sample can be stored at room temperature 30-90 minutes or 4 ºC up to six hours to prevent degradation of salivary molecules[31].”
Line 203: How did you calculate the volume of collected saliva? Weighing several empty Eppendorfs, and its average subtracts it with one with the saliva obtained? Or approximately with the value of the volume marked in the Eppendorf’s lateral? Please note that the volume value provided by the Eppendorf is only an estimate.
We estimated the volume of the collected saliva by using Eppendorph tubes with volume markers on the side. We have added description of this in the manuscript.
Line 217: The authors can refer to Table 4 to specify which biomarkers they refer to.
We added this to the manuscript (now Table 3).
Lines 222-223: Check the word “pERformed”. I do not understand the sentence “(…) and the operator of the assay was, blinded to if the sample was from a patient or healthy individual”. I advise removing it.
We removed this sentence according to your suggestions
Lines 224-225. Please, remove this sentence since you wrote it in line 219: “The MESO bioassay was performed according to the manufacturer’s specification and instruction.”. Change the sentence like the following: “The precisions based on both intra- and inter assay coefficient of variations (CV) were (…)”
We changed this sentence according to your suggestions
Lines 226 and 227: Please, write a point after the word “limits”. Write the following: “For more detail about the analytical validation from the inflammatory parameters measured, see table 4 and the supplementary table 1 [32.33].”
Changed according your suggestions. We have removed the supplementary table since the information can be found in Table 3.
Lines 229-231: This sentence could go better in a Discussion paragraph as a limitation of the study..
We agree that it will better fit in the Discussion section. Changed according to your suggestion.
Line 233: Give information about the intra and inter-assay coefficient of variation results, and detection limits offered by the manufacturer.
This information is added to the Materials and Methods section as well as in Table 3.
Line 235: Did you assess the normality distribution? Or it was decided because there was a big difference between both groups, the JIA and healthy group? IN any case, you must justify adequately why you used a non-parametric test. Change “25th/75th” for “25th-75th.
We have justified the use of non-parametric statistics in the Materials and methods section.
Lines 236-237: Complete the sentence: “For analytical statistics, the Mann-Whitney U-test was used to calculate the significant differences with respect to the time of collection, volume collected and flow rate between groups (patients vs. healthy individuals) or between the ten first or last samplings.”
We changed this in the manuscript
Results
Table 1. The units of volume are mL and not mL/min.
Thank you. We have corrected the table and removed the “Volume” since we sampled until we obtained sufficient volume, i.e. the volume is not really interesting here.
Line 86: How did you calculate it? You need to specify the kind of statistic calculation in the “Statistical analysis” sub-heading.
We have tried to clarify this under “Statistics”.
Lines 78-79: Those first and last ten samplings are only from the JIA group or form the healthy group or from both?
We have added this (important, thank you!) information to the Materials and Methods section.
Table 3. There is no point in offering the results of time collection, volume, and flow rate from these six healthy controls since these data must be as the previous one. You can describe only the results from the protein concentration.
We have removed Table 3 and added the protein concentration to the text.
Line 93: Were not 40 inflammatory biomarkers according to the information offered in the “Material and Methods” heading?
Sorry. Changed to 39, as it should be.
Discussion
Lines 104-109: Please, describe which and which not inflammatory mediators can be detected in every sample.
We have added a portion to the Discussion section.
Please, provide a paragraph of limitations. For example, no power study to determine if the statistic study had a type 2 error was performed, a higher proportion of girls than boys in both groups; and ideally, it would be desired also to assessing of the inflammatory mediators included in this study in saliva from the JIA patients.
Regarding power analyses and sex distribution, we have clarified these matters under Materials and Methods: “This methodological proof-of-concept study is part of a large longitudinal study of TMJ in JIA patients in relation to blood, saliva, periodontal and radiographic biomarkers. There are no previous studies on inflammatory markers in saliva in JIA children, nor are there any high-quality studies on prediction of TMJ involvement in JIA. A proper power calculation was therefore difficult, also considering that there are only a very limited number of JIA patients available. We included all JIA patients referred to the Center for Oral Rehabilitation from the four hospitals.”.
The biomarkers from all patients and healthy individuals included in this proof-of-concept study will be investigated in relation to clinical, radiological and blood variables in the next study. This study aimed to determine that our sampling method works.